# Assessment of the compound impact of sea level rise, land subsidence and storm surge under climate change in ShangHai

Bing Liang[1,2]*, Guoqing Shi[3]*, Yuexi Wu[4], Yuxuan Zhu[4], Mark Wang[2], Zhonggen Sun[3]

**1** Business School, Wuxi Taihu University, Wuxi, China, **2** Centre for contemporary Chinese studies, University of Melbourne,Parkville, Australia, **3** National Research Center for Resettlement, Hohai University, Nanjing, China, **4** College of Geography and Remote Sensing, Hohai University, Nanjing, China

* bliang1@126.com (BL); gshi@hhu.edu.cn (GS)

## Abstract

Global climate change-induced sea level rise has emerged as a critical environmental challenge for coastal cities in the 21st century. Shanghai, China's economic, financial, and shipping hub, faces significantly amplified inundation risks in its coastal areas due to the compounding effects of sea level rise, land subsidence, and storm surges. This study constructs a multi-case simulation framework using the sixth assessment report of intergovernmental panel on climate change sea level projection data, land subsidence monitoring records, and historical storm surge data to evaluate the impacts of three cases on future inundation risks: sea level rise alone (case1), sea level rise combined with land subsidence (case 2), and sea level rise coupled with land subsidence and storm surges (case 3). Leveraging Autoregressive Integrated Moving Average model time-series modeling, Geographic Information System spatial analysis, and numerical simulations, the study predicts relative sea level rise and inundation extents for 2050, 2070, and 2100. Results indicate that Shanghai's relative sea level rise rate far exceeds the global average, with land subsidence and storm surges synergistically amplifying disaster risks in low-lying coastal zones. Under case 1, the projected inundation area reaches 361.32 km$^2$ by 2100. Case 2 increases this area to 460.97 km$^2$, while case 3 shows a dramatic escalation to 1,331.91 km$^2$ by 2100—a surge of 870.94 km$^2$ compared to case 2—highlighting the dominant role of storm surges in extreme weather events. Spatial analysis identifies Chongming District, Pudong New Area, and Fengxian District as high-risk zones, with Chongming Island being the most severely affected (54.5% inundation by 2100). This study elucidates the compound impact mechanisms of sea level rise, land subsidence, and storm surges in Shanghai, providing a scientific foundation for coastal disaster mitigation and adaptive urban management. Recommendations include enhancing coastal flood defenses, optimizing land-use planning, improving extreme weather early-warning systems, and fostering international collaboration and technological innovation to bolster urban resilience against climate risks.

**Data availability statement:** All relevant data are within the manuscript and its Supporting Information files. Data Registration DOI: https://doi.org/10.17605/OSF.IO/V6DRJ.

**Funding:** This research was supported by the National Social Science Fund of China (23CXW034).

**Competing interests:** The authors declare no conflict of interest.

## 1. Introduction

Global climate change-induced sea level rise (SLR) has become one of the most critical environmental challenges of the 21st century, posing severe threats to the sustainable development of coastal cities worldwide [1,2]. The Intergovernmental Panel on Climate Change (IPCC) Sixth Assessment Report (AR6) indicates that under global warming, global sea levels will continue to rise in the coming decades, In the IPCC AR6, Shared Socioeconomic Pathways (SSP) are internally consistent socioeconomic, emissions, and climate projections that explore five alternative futures based on varying socio-economic assumptions, geopolitical trends, and mitigation/adaptation efforts. The AR6 emphasizes that confidence levels vary by variable, region, and scenario, with stronger consistency in global-scale projections (e.g., temperature) than in regional extremes or long-term feedbacks. [3]. Against this backdrop, coastal regions face significant challenges to ecological security, socioeconomic stability, and infrastructure resilience [4]. However, coastal cities are not only threatened by SLR but also by the compound effects of land subsidence and storm surges [5,6]. These three factors interact synergistically, creating a "rising seas and sinking land" phenomenon that significantly amplifies disaster risks in coastal urban areas.

Land subsidence represents another critical challenge for coastal cities, particularly in regions with complex geological conditions, excessive groundwater extraction, and rapid urbanization [7]. Cities such as Shanghai, Tokyo, and Jakarta have experienced pronounced subsidence in recent years, accelerating relative SLR rates far beyond the global average and exacerbating risks of flooding and salinization in coastal zones [8]. Furthermore, storm surges—intensified by global warming and SLR—are increasing in both frequency and severity, further jeopardizing coastal safety [9].

Despite extensive research on SLR, land subsidence, and storm surges individually, current studies predominantly focus on single-factor analyses, with limited systematic evaluation of their compound impacts. For instance, studies such as those by Rahmstorf (2010) have highlighted significant regional disparities in SLR impacts on coastal zones, emphasizing disproportionate threats to cities in East Asia and Southeast Asia [10]. Research on land subsidence, like that conducted in Jakarta by Abidin et al. (2011), has documented subsidence rates as high as 25 cm/yr, far surpassing global SLR rates [11]. Furthermore, studies such as Kirezci et al. (2020) have projected that under high-emissions scenarios, storm surge-induced flooding could expand by nearly 50% globally by 2100, underscoring the increasing frequency and severity of storm surges [12]. However, these studies primarily analyze single factors, with fewer investigations systematically evaluating the compound effects of SLR, land subsidence, and storm surges, as highlighted in the present study.This study aims to address this gap by integrating multi-source remote sensing data, climate model projections, and numerical simulations to comprehensively assess the combined effects of SLR, land subsidence, and storm surges. It further investigates their synergistic mechanisms in driving coastal disaster risks. The findings are intended to provide scientific foundations for disaster mitigation and adaptive policymaking in coastal cities, offering critical decision-support for global climate resilience efforts.

## 2. Literature review

The primary drivers of SLR include thermal expansion, ice sheet melting, and glacier retreat [13,14,15]. Recent studies highlight significant regional disparities in SLR impacts on coastal zones [16]. For instance, Rahmstorf [17]emphasized that 21st-century SLR will disproportionately threaten cities in East Asia, Southeast Asia, and the eastern coast of North America. In China, regions such as the Yangtze River Delta, Pearl River Delta, and Bohai Bay exhibit relative SLR rates exceeding the global average, partly due to regional climatic and hydrological dynamics [18].

Land subsidence, widely documented in coastal cities globally, has emerged as a critical amplifier of SLR impacts [19]. Jakarta, for example, experiences subsidence rates as high as 25 cm/yr—far surpassing global SLR rates [11]. Chinese coastal cities like Shanghai and Tianjin also face persistent subsidence driven by groundwater extraction and soft soil compaction [20]. Research indicates that subsidence exacerbates coastal flooding risks and compromises the efficacy of flood defense systems [21], underscoring the need to integrate subsidence into relative sea level assessments for coastal risk management.

Storm surge frequency and intensity have increased globally, particularly in the North Atlantic and western Pacific. Studies demonstrate that SLR amplifies storm surge impacts, rendering previously secure coastal areas increasingly vulnerable to extreme events [22]. For example, Kirezci [12]projected that under the RCP8.5 high-emissions scenario, storm surge-induced flooding could expand by nearly 50% globally by 2100. Furthermore, the interplay between storm surges and subsidence magnifies disaster risks, as evidenced by Hurricane Sandy's unprecedented flooding in New York (2012), where cumulative relative SLR and subsidence exacerbated impacts.

While existing research elucidates the individual effects of SLR, subsidence, and storm surges, systematic evaluations of their compound impacts remain limited. Moreover, most studies rely on historical regression analyses, with few projecting future scenarios. Addressing these gaps, this study contributes by: Developing a multi-scenario assessment framework: Integrating multi-source remote sensing data, climate models, and numerical simulations to quantify the compound effects of SLR, subsidence, and storm surges on coastal disaster risks.Focusing on Shanghai as a case study, leveraging long-term observational data to analyze spatiotemporal patterns of these factors and assess their implications for urban resilience.Unlike prior regional compound-impact studies that often rely on historical regression analyses, this research develops a multi-scenario assessment framework that integrates multi-source remote sensing, climate models, and numerical simulations to project future risks. Furthermore, it uniquely quantifies the spatiotemporal interplay of relative sea-level rise, incorporating both climate-driven SLR and localized land subsidence, with storm surges. This approach provides a forward-looking, high-resolution analysis of compound hazards specifically for Shanghai, leveraging long-term observational data to offer actionable insights for urban resilience.This research not only advances understanding of compound climate-hazard interactions but also provides actionable insights for adaptive coastal governance, offering policy-relevant guidance for global sustainable development efforts.

## 3 Methods

### 3.1 Study area

Shanghai, located in eastern China (120°51′E–122°12′E, 30°40′N–31°53′N), serves as a vital coastal gateway city. Situated on the western Pacific coast near the Yangtze River estuary, it lies at the confluence of the Yangtze River and the East China Sea. Bordered by the East China Sea to the east, Hangzhou Bay to the south, and Jiangsu and Zhejiang provinces to the west, Shanghai anchors China's most economically developed Yangtze River Delta region. Its strategic location has established the city as a global economic, financial, trade, and shipping hub.

Shanghai's terrain is predominantly flat, forming part of the Yangtze River Delta alluvial plain, with an average elevation of approximately 2.19 meters. The land slopes gently from east to west. While low-lying plains dominate, the southwestern area features minor hills and mountains, including Dajinshan—the highest natural point at 103.4 meters above sea level.

Although the flat terrain supports agriculture and urban expansion, it also exacerbates geological hazards such as land subsidence and flooding, particularly under escalating SLR and climate extremes [23].

Shanghai experiences a subtropical monsoon climate, characterized by distinct seasons, mild temperatures, and abundant humidity. The annual average temperature is around 15.8°C, with precipitation exceeding 1,200 millimeters, concentrated during the plum rain season. Typhoons and heavy rainfall, prevalent during Shanghai's spring and summer seasons, impose significant strains on urban infrastructure, exacerbating vulnerabilities in drainage systems, flood defenses, and transportation networks.Intense precipitation often overwhelms stormwater drainage capacities, leading to urban flooding, waterlogging, and potential damage to underground utilities [24]. Simultaneously, typhoon-induced storm surges elevate coastal water levels, increasing hydrostatic pressure on seawalls and tidal barriers, which may result in structural failures if design thresholds are exceeded [25]. The city's coastal location subjects it to complex hydrological conditions influenced by the East China Sea and Yangtze River. The dense river network, including the Huangpu and Wusong Rivers, not only provides abundant water resources and navigation convenience but also shapes a unique waterscape.

Administratively, Shanghai comprises urban districts and three offshore islands: Chongming, Changxing, and Hengsha. Chongming Island, China's third-largest island, holds critical ecological significance. As a coastal metropolis, Shanghai has abundant coastline resources. Its coastal wetlands and estuary areas play a crucial role in environmental protection. Chongming Island's wetlands, designated as one of the globally important wetland reserves, are especially significant for bird migration and ecological balance [26].

### 3.2 Data sources

① SLR projections: Derived from the IPCC AR6 dataset accessed via the National Aeronautics and Space Administrat's (NASA) Sea Level Change Portal.

Since the IPCC AR6 has already produced regional sea-level projections,this study adopts these projected values for the following research. The IPCC AR6 provides projections of sea level for five Shared Socioeconomic Pathways (SSP) including SSP1–1.9,SSP1–2.6, SSP2–4.5, SSP3–7.0 and SSP5–8.5 based on the average sea level from 1995 to 2014. These scenarios depict various different development patterns of future economic and social systems, reflecting the correlation between economic and social development and the challenges of mitigating and adapting to climate change. They are the core foundation for conducting climate change impact assessments and climate policy formulation.To highlight the research theme, which is the combined impact of SLR, land subsidence, and storm surges on coastal cities, and to reduce computational workload, we use sealevel projection data under the emission scenario SSP2–4.5 for analysis. The projected data was downloaded from the NASA Sea Level Change Portal.

The name of the tide gauge station we selected is "LUSI," which is located in China. The station is situated in Lvsigang Town, Rudong County, Jiangsu Province, and is one of the important tide gauge stations along the Yellow Sea coast, frequently used for research on sea level changes in China's coastal waters. The query website is: https://sealevel.nasa.gov/ipcc-ar6-sea-level-projection-tool?psmsl_id=979&data_layer=scenario.

② Land subsidence data: Obtained from the Shanghai Geological Environment Bulletin (2009–2023)(https://hd.ghzyj.sh.gov.cn/dzkc/dzhjbg/202504/t20250410_1215068.html). The relevant departments in Shanghai will regularly measure and publish ground subsidence data.

The land subsidence data used in our study and the Vertical Land Motion (VLM) data available on the NASA portal are fundamentally different in source and dimension. Our analysis relies on official, ground-based monitoring data published annually in the Shanghai Geological Environment Bulletin. This data is derived from a network of leveling benchmarks and borehole extensometers specifically designed to measure compaction of the shallow soil layers, which is the primary driver of subsidence in Shanghai due to groundwater withdrawal and construction load. In contrast, the VLM data on the NASA portal is typically derived from satellite-based remote sensing, which measures displacement of the land surface and can capture different signals. We used the NASA portal solely for the climate-induced sea-level componentand not for

the VLM data, as the local, high-resolution official dataset is more directly relevant and accurate for assessing the specific anthropogenic subsidence processes in Shanghai.

### 3.3 Methodology

**3.3.1 Case design.** SLR is divided into 3 cases, which describe changes based on different benchmarks. Case 1 considers only global SLR; Case 2 examines relative SLR incorporating land subsidence; Case 3 evaluates relative SLR under the combined impacts of land subsidence and storm surges. See Fig 1 for details. Global SLR is affected only by global factors, while relative SLR is influenced by both global and local factors [27]. Therefore, this study incorporates local factors like land subsidence and storm surges when examining SLR in Shanghai.

To account for localized impacts, this study evaluates three cases for Shanghai:

Case 1: Global SLR only.

Case 2: Global SLR combined with land subsidence.

Case 3: Global SLR combined with both land subsidence and storm surges.

**3.3.2 Sea level rise projection methodology.** The SLR projections for Shanghai (2050, 2070, 2100) were derived using projections data from the IPCC AR6 report and the NASA Sea Level Change Portal. On the portal, the projections

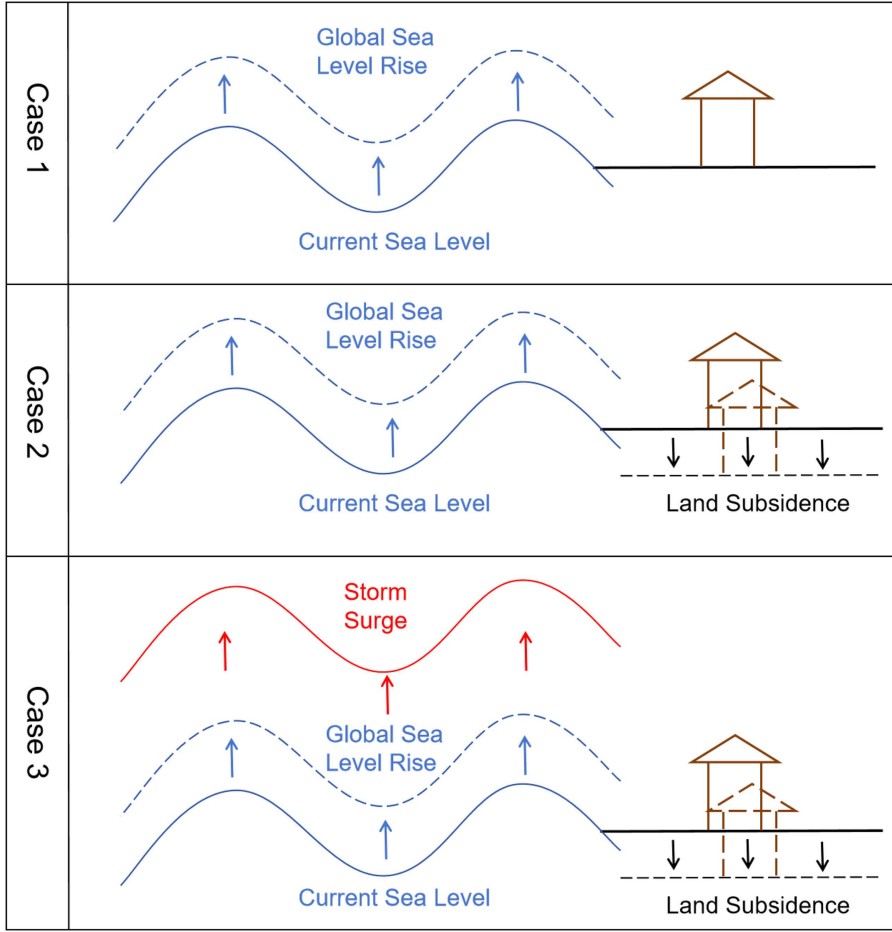

**Fig 1. Schematic diagram of different case modes.**

data from the nearest coastal station to Shanghai was used as the basis for Shanghai's SLR projections. Under the SSP2–4.5 scenario, SLR projections for Shanghai were obtained for the years 2050, 2070, and 2100.

**3.3.3 Land subsidence prediction methodology.** Time series analysis is a statistical, mathematical, and econometric technique for processing sequential data to identify patterns, trends, and periodicities, enabling forecasting and inference [28]. Among various methods, the autoregressive integrated moving average (ARIMA) model is a widely adopted classical approach for time series analysis. This model integrates three core components: autoregression (AR), differencing (I), and moving average (MA), making it suitable for analyzing diverse types of time series data.The projection is intended to highlight the potential long-term cumulative effect if current subsidence mechanisms persist without mitigation. We fully agree that this approach has limitations. The projection does not account for potential future nonlinearities, changes in groundwater management policy, or other mitigating interventions that could alter the subsidence rate.

The ARIMA model converts non-stationary time series into stationary ones through differencing, then models and forecasts the data using AR and MA components. The general form of an ARIMA model is denoted as ARIMA(p, d, q), where: p: Number of autoregressive terms. d: Order of differencing to achieve stationarity. q: Number of moving average terms. The AR component captures the linear relationship between a variable's current value and its past values. The differencing (I) component addresses non-stationarity by removing trends and seasonality. The MA component refines predictions using a weighted average of past forecast errors. The formula is in the form of:

$$y_t = \mu + \sum_{i=1}^{p} \gamma_i y_{t-i} + \in_t$$

(1)

Where $y_t$ is the current value, $\mu$ is a constant, $\gamma_i$ are autoregressive coefficients, and $\in_t$ is the error term.

Differencing (I): A critical step to stabilize non-stationary data involving d times of differencing.

MA model: Focuses on the accumulation of errors in autoregressive models. It forecasts current values via a linear combination of past white – noise errors. The formula is:

$$y_t = \mu + \sum_{i=1}^{q} \theta_i \in_{t-i} + \in_t$$

(2)

where $\theta_i$ are moving average coefficients.

In this study, the ARIMA model predicts land – subsidence in Shanghai. As data is available from 2009–2023, the land subsidence data is extrapolated backwards to 1995 to align with the absolute sea – level rise baseline. Then, data from 1995–2023 is used to forecast subsidence from 2024–2100. Finally, using 1995–2014 as the baseline, the cumulative subsidence is projected to.

We acknowledge that projecting subsidence to 2100 based on a limited historical record is a significant challenge and a source of uncertainty. The ARIMA model was applied to extrapolate the current linear trend as a first-order approximation. This projection is not intended to predict nonlinear future changes but to provide a plausible scenario if the dominant mechanisms of subsidence (e.g., current groundwater practices, urban load) persist without major mitigation interventions. The results, therefore, highlight the potential cumulative impact under a "business-as-usual" assumption.

**3.3.4 Storm surge inundation delineation.** The inundation width (D) resulting from storm surges height (H) and nearshore average slope (θ):

$$D = H/\tan \theta$$

(3)

Where H is the storm surge height, θ is the nearshore average slope, D is the inundation width.

In this study, the slope of Shanghai was calculated using the city's digital elevation model(DEM) data and the ArcGIS Pro platform. The average slope within 2 km of the coastline was derived and used as the near – shore average slope. After determining the value of H, the value of D was calculated. All these processes were implemented in ArcGIS Pro for visualization.

**3.3.5 Inundation area calculation methodology.** Using the ArcGIS Pro platform, the collected DEM data for Shanghai were reclassified according to the projected SLR values. The reclassified raster data were then converted into vector polygons through a "raster-to-polygon" operation. An area field was added to the attribute table, and geometric calculations were performed to derive the inundation area under each SLR case.

# 4. Results

The results presented from our current method are interpreted cautiously as a central estimate scenario rather than a full probabilistic assessment. The values for these cases are central estimates based on the median SLR and median subsidence projection.As our study utilizes the standard projections from the IPCC AR6 Sea Level Projection Tool, which include processes related to ice-sheet instability (e.g., marine ice sheet instability and structural failure of ice cliffs), the appropriate confidence level for the projections under SSP2–4.5 is medium Confidence.

## 4.1 Global sea level rise projections

The "global sea level" rise we refer to here refers to the SLR without considering the effects of land subsidence and storm surges. If we consider the effects of land subsidence and storm surges, then SLR at this point refers to "relative SLR". Based on IPCC AR6 projections, the predicted absolute SLR for Shanghai is summarized in Table 1.

The projected global SLR for Shanghai under the SSP2–4.5 scenario reflects mid-range climate projections, yet it still poses significant threats due to the city's low-lying topography. Compared to global averages, Shanghai's exposure is exacerbated by regional hydrological and climatic dynamics, aligning with broader findings that East Asian coastal zones face disproportionate risks from climate-induced SLR.

## 4.2 Land subsidence projections

The predicted values of relative SLR rise after incorporating land subsidence data are shown in Table 2, with the baseline period (1995–2014) aligned with global SLR reference years.

Land subsidence in Shanghai, driven largely by anthropogenic factors such as groundwater extraction and urban load, significantly accelerates relative SLR, compounding flood risks beyond those caused by climatic factors alone. This aligns

**Table 1. Projections of relative sea level rise in Shanghai (unit: m).**

| Year | Uncertainty Interval (17th–83rd percentile) | Global sea level rise |
|------|---------------------------------------------|------------------------|
| 2050 | 0.26-0.40 | 0.32 |
| 2070 | 0.40-0.63 | 0.49 |
| 2100 | 0.63-1.05 | 0.81 |

**Table 2. Prediction of land subsidence in Shanghai (unit: m).**

| Year | Uncertainty Interval (95% Prediction Interval) | Accumulation of land subsidence |
|------|------------------------------------------------|----------------------------------|
| 2050年 | 0.22-0.34 | 0.28 |
| 2070年 | 0.28-0.42 | 0.35 |
| 2100年 | 0.34-0.52 | 0.43 |

with studies in cities like Jakarta and Tokyo, where localized subsidence dramatically amplifies coastal vulnerability, highlighting the critical role of human activity in shaping regional exposure.

## 4.3  Relative sea level rise projections

By integrating land subsidence data, the projected relative SLR values for Shanghai are detailed in Table 3.

The integration of global SLR and local land subsidence reveals that relative SLR in Shanghai far exceeds global averages, illustrating a synergistic mechanism where natural and human-induced factors jointly elevate inundation risks. This compounded effect underscores the necessity of incorporating localized subsidence into risk assessments, as emphasized in recent studies on sinking delta cities.

## 4.4  Storm surge inundation projections

Using historical extremes from the Huangpu Park Station (peak storm surge height: 5.44 m) [29], this study adopted a conservative surge height (H) of 6 m accounts for potential future intensification of storm surges under climate change and provides a conservative estimate for inundation modeling. The nearshore average slope ($\alpha\alpha$) was calculated as 0.12° (tangent ≈ 0.0021), yielding a storm surge inundation width (D) of approximately 2,857.14 m via the formula $D = H/\alpha D = H/\alpha$. ArcGIS Pro simulations for Case 3 produced the inundation area projections listed in Table 4.

Storm surges introduce extreme, episodic inundation risks that dominate overall hazard projections, with simulated impacts indicating widespread flooding particularly in low-lying districts like Chongming and Pudong. This aligns with

**Table 3.  Projections value of relative sea level rise in Shanghai (unit: m).**

| Year | Global sea level rise | Accumulation of land subsidence | Relative sea level rise |
|------|----------------------|----------------------------------|-------------------------|
| 2050 | 0.32 | 0.28 | 0.6 |
| 2070 | 0.49 | 0.35 | 0.84 |
| 2100 | 0.81 | 0.43 | 1.24 |

**Table 4.  Predicted inundation area of Shanghai in case 3 (unit: km².).**

| Area | 2050 | 2070 | 2100 |
|------|------|------|------|
| Xuhui District | 0.00 | 0.00 | 0.00 |
| Jiading District | 0.00 | 0.00 | 0.00 |
| Songjiang District | 0.00 | 0.00 | 0.00 |
| Qingpu District | 0.00 | 0.00 | 0.00 |
| Putuo District | 0.00 | 0.00 | 0.00 |
| Minhang District | 0.00 | 0.00 | 0.00 |
| Jinshan District | 47.15 | 47.15 | 55.42 |
| Chongming District | 697.02 | 697.02 | 725.28 |
| Baoshan District | 48.07 | 48.07 | 55.69 |
| Pudong New Area | 387.95 | 387.95 | 393.17 |
| Yangpu District | 0.00 | 0.00 | 0.00 |
| Hongkou District | 0.00 | 0.00 | 0.00 |
| Jing'an District | 0.00 | 0.00 | 0.00 |
| Huangpu District | 0.00 | 0.00 | 0.00 |
| Fengxian District | 100.88 | 100.88 | 102.35 |
| Changning District | 0.00 | 0.00 | 0.00 |
| Total | 1281.07 | 1281.07 | 1331.91 |

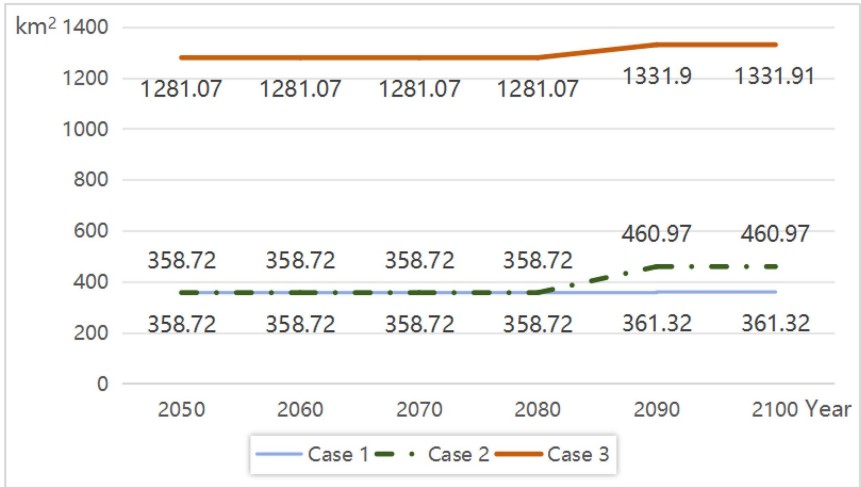

global research showing that storm surges, when coupled with SLR and subsidence, exponentially increase disaster extent and severity, underscoring the need for enhanced early-warning and infrastructure resilience.

## 4.5 Inundation area projections across cases

Three cases were simulated to assess future inundation trends in Shanghai for 2050, 2070, and 2100. Notably, inundation primarily expanded across eastern coastal zones, including Pudong New Area, Chongming District, and Fengxian District. Low-lying regions such as Chongming Dongtan (East Chongming Wetlands) and Nanhui Coastal Area were disproportionately impacted.

From Table 5 and Fig 2, inundation areas under all three cases exhibit an increasing trend over time, with Case 3 (SLR + subsidence + storm surges) far exceeding the impacts of Cases 1 and 2. In 2050, inundation area under Case 1 (SLR alone) is 358.72 km$^2$, rising to 361.32 km$^2$ by 2100—a marginal increase. With subsidence added (Case 2), the inundation area reaches 460.97 km$^2$ by 2100, marking a 27.6% increase compared to Case 1, highlighting the cumulative amplifying role of subsidence. Most strikingly, Case 3 (including storm surges) causes inundation areas to surge from 1,281.07 km$^2$ in 2050–1,331.91 km$^2$ by 2100—a staggering increase of 870.94 km$^2$ over Case 2, underscoring the critical threat of storm surges under future extremes.

(1) Case 1 (SLR Only)

From 2050 to 2100, inundation areas remain relatively stable (~360 km$^2$), concentrated in eastern Chongming District, coastal Fengxian District, and low-lying zones of Pudong New Area. While short-term impacts (2050–2070) are limited, gradual erosion in critical regions necessitates long-term monitoring (by 2100).

**Table 5. Predicted value of inundation area under three cases (unit: km$^2$).**

| Case | 2050 | 2060 | 2070 | 2080 | 2090 | 2100 |
|---|---|---|---|---|---|---|
| Case 1 | 358.72 | 358.72 | 358.72 | 358.72 | 361.32 | 361.32 |
| Case 2 | 358.72 | 358.72 | 358.72 | 358.72 | 460.97 | 460.97 |
| Case 3 | 1281.07 | 1281.07 | 1281.07 | 1281.07 | 1331.90 | 1331.91 |

**Fig 2. Prediction of inundation area under three cases (2050-2100).**

(2)  Case 2 (SLR + Subsidence)

Post-2070, subsidence effects intensify, adding ~102.25 km² of inundation by 2100, primarily expanding into Pudong New Area, Baoshan District, and Fengxian District. Cumulative subsidence exacerbates relative SLR, increasing long-term flood retention risks even without extreme weather.

(3)  Case 3 (SLR + Subsidence + Storm Surges)

Storm surges dominate, driving inundation to 1,281.07 km² by 2050 and 1,331.91 km² by 2100–870.94 km² higher than Case2. Table 4 reveals extensive coastal inundation, particularly in Chongming Island, Jinshan coastal zones, and industrial areas of Pudong. Inland expansion threatens riverside urban districts (e.g., Yangpu, Hongkou) under extreme conditions, exposing critical infrastructure (ports, industrial parks) to severe risks. These results indicate that Shanghai's existing seawalls may prove inadequate in withstanding extreme storm surge events in the future

## 5. Discussion

### 5.1 Compound effects of SLR and subsidence amplify inundation risks

Studies show that Shanghai, a coastal low – lying city, faces increased coastal flooding risks due to SLR rise caused by global warming and land subsidence induced by long-term human activities, such as groundwater extraction [30]. The combined effect of these two factors is not simply linear but exhibits a nonlinear amplification effect, particularly during extreme weather events, the extent and depth of flooding exceed expectations.The absence of quantified uncertainty intervals or confidence levels for the sea-level rise, subsidence, and inundation projections limits the ability to fully assess the range of potential future risks. This uncertainty, stemming from climate model variability, subsidence rate assumptions, and storm surge intensity, means that the presented inundation areas should be considered central estimates rather than definitive outcomes. Consequently, adaptive policy and infrastructure planning must incorporate this inherent uncertainty by preparing for a spectrum of possible scenarios to ensure robust and resilient coastal management.

**5.1.1  Mechanisms of compound impacts.**  SLR is mainly driven by thermal expansion and ice – sheet melting due to global warming [31], while land subsidence is more influenced by regional factors such as excessive groundwater extraction, geological activity, and urbanization. However, it can also be influenced by global-scale processes such as Glacial Isostatic Adjustment (GIA) and Gravity, rotation, and deformation (GRD)effects, which modify land elevation over broader spatial and temporal scales. When both occur simultaneously, the rate of relative SLR far surpasses the global average [32]. For instance, in this study's projection under the IPCC SSP2–4.5 scenario, Shanghai's global SLR by 2100 will reach 0.81 m, with about 0.43 m of cumulative land subsidence in the same period. After combining the two, the actual relative SLR will be as high as 1.24 m. This indicates that even in the absence of additional extreme weather events, the long-term flooding risk in low-lying areas of Shanghai will significantly increase.

Moreover, land subsidence not only directly lowers the land surface but also undermines the stability of foundation structures, weakening the stability of urban flood – prevention and drainage systems [33]. For example, ground subsidence can deform underground pipelines, reducing drainage capacity and worsening urban flooding during heavy rains. During storm surges, the lower land height allows tides to infiltrate further inland, expanding the flooded area and prolonging the duration of floodwater retention [34].

**5.1.2  Spatial heterogeneity.**  In Shanghai, varying ground – subsidence rates across districts lead to different relative sea – level rise impacts [35]. Pudong, Baoshan, and Chongming Districts, near the coast with soft soil, have high subsidence rates, worsening flood risks. By 2100, Chongming District's flooded area may reach 725.28 km², 54.5% of the city's total. Inland districts like Jing'an and Huangpu, elevated with slow subsidence, are less affected. Land use also affects flood risk [36]. Some industrial and residential zones, low – lying due to land – reclamation or foundation sinking, are more vulnerable to sea – level rise. For example, Pudong's coastal industrial parks face both subsidence and tidal

threats, while Chongming Island, though with a lower subsidence rate, may still see significant impacts on farmland and the ecosystem due to its low elevation.

**5.1.3 Cumulative effects and their impacts.** The compounding effects of SLR and land subsidence not only lead to long-term land loss but also profoundly impact Shanghai's infrastructure, ecological environment, and socioeconomic development.

(1) Infrastructure Vulnerabilities

The rise in relative sea level exposes coastal defenses (such as seawalls and tidal gates) to increased hydrostatic pressure, potentially exceeding their design thresholds and causing failure [4,37]. For instance, some of Shanghai's tidal barriers were designed based on sea level standards from 50 years ago.If the relative SLR by more than 1.2 meters in the future, the original flood defense capability will be significantly compromised. Additionally, Land subsidence may cause structural deformation of infrastructure [38]. Especially in areas with deep soft soil foundations [39], such as certain metro lines in Pudong and Yangpu, which may face the risk of tunnel deformation due to foundation settlement

(2) Ecological Degradation

Saltwater intrusion intensifies the salinization of coastal wetlands, impairing the health of marine ecosystems. Furthermore, land subsidence alters the flow paths of river networks, degrading water quality [33].Coastal wetlands and estuarine ecosystems, such as the Chongming Dongtan Wetland—a critical stopover site on the East Asian-Australasian Flyway for migratory birds—are under increasing pressure due to the combined effects of sea-level rise and land subsidence, which have caused significant wetland area reduction and subsequent degradation of ecosystem services [40,41,42].

(3) Socioeconomic Losses

Land inundation and infrastructure damage directly impact residents' lives and economic activities [43,44,45]. By 2100, economic losses in Shanghai's coastal industrial zones, logistics parks, and residential areas could reach hundreds of billions of yuan [46], with exorbitant relocation and reconstruction costs. Real estate markets may also suffer, as property values in low-lying coastal areas decline due to chronic flood risks [47]. Some residents may have to move, which can cause socioeconomic issues like housing shortages and employment pressures [48,49].

The combined effects of SLR and land subsidence significantly amplify Shanghai's flooding risk [7], especially during extreme events such as storm surges, when the disaster impacts are even more severe. Urban planning and disaster management need to comprehensively consider the long-term impacts of both, and adopt proactive adaptive measures to enhance the city's resilience to climate change.

## 5.2 Storm surges intensify the destructive power of compound disasters

In the context of global climate change, Shanghai faces multifaceted and complex disaster threats from SLR, land subsidence, and storm surges. Among these, storm surges—extreme weather events characterized by increasing frequency and intensity—significantly amplify disaster risks when compounded with SLR and subsidence [50].

**5.2.1 Synergistic effects of storm surges and sea level rise.** Storm surges, triggered by intense weather systems such as typhoons and extratropical cyclones, are abrupt, high-intensity phenomena with widespread impacts [51]. Under global warming, rising sea surface temperatures and increased atmospheric moisture create favorable conditions for storm surge formation [52]. Climate change not only escalates the frequency and intensity of extreme weather events but also indirectly exacerbates storm surge threats by altering ocean circulation and atmospheric dynamics.

The synergistic effects of storm surges and SLR are particularly pronounced. SLR, a direct consequence of climate change, elevates the baseline water level for storm surges [1]. This means that even in the absence of storm surges, low-lying coastal areas will face heightened inundation risks. When storm surges occur, SLR further intensifies their destructive power, enabling tidal waters to more easily breach seawalls, expand inundation extents, and increase flooding

depths [53]. Projections indicate that by 2100, Shanghai's global SLR will reach 0.81 meters, while relative SLR (incorporating land subsidence) will rise to 1.24 meters. This upward trend in sea levels will undeniably magnify the threat of storm surges to Shanghai.

**5.2.2 Compounding impacts of storm surges and land subsidence.** Land subsidence, driven by excessive groundwater extraction, compression of soft soil foundations, and rapid urbanization [40], poses another critical challenge for Shanghai. Subsidence lowers terrain elevation, rendering coastal areas more vulnerable to storm surges [54]. As shown in Table 2, cumulative subsidence in Shanghai is projected to reach 0.43 meters by 2100. This subsidence not only directly reduces land elevation but also destabilizes urban infrastructure, weakening its capacity to withstand storm surges.

When storm surges interact with SLR and subsidence, their destructive potential is significantly increased. Subsidence accelerates the rate of relative SLR, making low-lying coastal zones more susceptible to storm surge impacts [55]. Additionally, subsidence-lowered terrain reduces natural buffering against tidal intrusion, prolonging flood retention and expanding inundation areas. Furthermore, subsidence undermines the structural stability of coastal defenses (e.g., seawalls), diminishing their protective efficacy [56]. This compound interaction mechanism significantly elevates disaster risks for Shanghai during storm surge events.

**5.2.3 Regularity analysis of storm surge impacts.** Through an in-depth analysis of the data tables presented in this study, we further elucidate the regularity patterns of storm surge impacts. Temporally, as SLR and land subsidence persist, storm surge-induced inundation risks for Shanghai will continue to escalate. As shown in Table 5, under Case 3 (SLR + subsidence + storm surges), the projected inundation area in Shanghai exhibits a year-by-year increasing trend from 2050 to 2100. This trend reflects the long-term and cumulative nature of interactions between storm surges, climate change, and land subsidence. Additionally, storm surge impacts demonstrate periodic characteristics, with frequency and intensity peaking during monsoon seasons and typhoon-active periods. Such periodicity implies heightened disaster risks for Shanghai during specific intervals, making it necessary to intensified weather monitoring, early warnings, and preventive measures to mitigate losses. Policy interventions, such as regulations on groundwater extraction and urban development planning, can directly mitigate anthropogenic drivers of land subsidence. Effective governance and targeted policies are crucial for slowing subsidence rates, thereby reducing future compound inundation risks [57].

The interplay of storm surges, SLR, and land subsidence not only exacerbates disaster risks but also profoundly impacts Shanghai's socioeconomic development and ecological environment [58]. Inundation of low-lying coastal areas due to SLR and subsidence directly disrupts livelihoods and economic activities, leading to massive financial losses [2]. Prolonged flooding and expanded inundation zones may also cause irreversible damage to coastal wetlands and estuarine ecosystems, threatening biodiversity and ecological balance [26].

Under climate change, the destructive power of storm surges in Shanghai has intensified significantly. Their compound effects with SLR and subsidence continually elevate coastal disaster risks. A thorough understanding of storm surge dynamics, compounding impacts, and regularity patterns is critical for formulating effective disaster prevention and mitigation strategies, ensuring Shanghai's sustainable development.

## 5.3 Spatial heterogeneity significantly impacts composite risk distribution

Under global climate change, Shanghai faces complex disaster risks from SLR, land subsidence, and storm surges [59]. These hazards not only independently threaten urban safety and development but also interact to create a compounding "rising seas and sinking land" effect, significantly amplifying disaster risks. As a vast, topographically diverse, and socioeconomically concentrated international metropolis, Shanghai exhibits pronounced spatial heterogeneity in its exposure to these risks across different regions. This heterogeneity is reflected not only in natural geographic conditions but also in socioeconomic structures, disaster resilience capacities, and ecological conditions.

**5.3.1 Spatial heterogeneity in natural geographic conditions.** Shanghai is situated on the Yangtze River Delta alluvial plain, characterized by overall flat terrain but significant internal topographic variations. Eastern coastal areas, particularly Chongming Island and Pudong New Area, are low-lying and vulnerable to direct impacts from SLR and storm surges. In contrast, western and southern districts such as Qingpu and Songjiang, though also part of the plain, have relatively higher elevations and face lower immediate threats from SLR. Additionally, elevated regions like Dajinshan (103.4 meters) act as natural "safe islands," offering better resistance to floods and storm surges.

Land subsidence, another critical hazard, also displays marked spatial heterogeneity. Driven by excessive groundwater extraction, soft soil compression, and rapid urbanization, subsidence rates vary significantly across Shanghai [60]. Coastal zones like Pudong New Area and Baoshan District, with complex geology and intensive groundwater use, experience higher subsidence rates (e.g., 5–10 mm/year), exacerbating their disaster risks. Inland areas, with lower groundwater extraction and subsidence rates, remain comparatively safer.

**5.3.2 Spatial heterogeneity in socioeconomic activities.** As China's economic, financial, trade, and shipping hub, Shanghai's socioeconomic activities are highly concentrated, with stark regional disparities in development levels and industrial structures. Coastal regions such as Pudong and Baoshan, benefiting from strategic locations and transportation networks, serve as vital industrial and logistics hubs. However, these areas face heightened disaster risks—floods or storm surges could cripple local economies.In contrast to coastal areas, inland regions in Shanghai have relatively lower economic development levels but more diversified industrial structures, resulting in smaller impacts from disasters. For instance, Qingpu and Songjiang Districts, with their focus on agriculture and light industry, exhibit stronger resilience against natural disasters. These areas can also achieve sustainable economic development by fostering emerging industries such as eco-tourism and cultural creativity.

**5.3.3 Spatial heterogeneity in disaster resilience.** Disaster resilience is a critical indicator of a city's capacity to respond to natural hazards. While Shanghai has achieved notable progress in disaster prevention and mitigation, significant disparities in resilience persist across its regions. Coastal areas, facing higher disaster risks, receive greater attention and investment from the government and stakeholders [61]. These zones prioritize infrastructure such as flood embankments and tidal gates, resulting in relatively stronger disaster resilience. In contrast, inland areas, with lower perceived risks, receive less focus and resources, leaving them underprepared for natural disasters. Moving forward, efforts in disaster management must emphasize regional balance and coordination, enhancing resilience in inland areas through targeted investments and capacity-building initiatives.

**5.3.4 Spatial heterogeneity in ecological conditions.** Ecological conditions serve as a crucial indicator of a city's sustainable development potentia. Shanghai's ecological status varies markedly across regions. Coastal areas, subjected to long-term stressors like SLR, storm surges, and subsidence, exhibit fragile ecosystems. The wetlands, estuaries, and other ecosystems in these areas have suffered severe damage, leading to a decline in biodiversity and weakened ecosystem services. Inland regions, however, boast more stable ecosystems and higher biodiversity. By strengthening ecological conservation and restoration efforts, these areas can enhance ecosystem resilience, providing robust support for the city's sustainable development.

**5.3.5 The comprehensive effect of spatial heterogeneity on the distribution of composite risk.** The impact of spatial heterogeneity on compound risk distribution is multi – faceted. First, the spatial heterogeneity of natural geographical conditions determines the degree of vulnerability of different regions to hazards such as SLR, land subsidence, and storm surges. Coastal areas, characterized by their low-lying topography and high rates of land subsidence, are prone to high disaster risks [62].Second, the spatial heterogeneity of socio – economic activities exacerbates the uneven distribution of disaster risks. As key economic hubs of Shanghai, coastal areas would suffer huge economic impacts if disasters struck. In contrast, inland areas, with relatively lower levels of economic development but diverse industrial structures, are less impacted by disasters.Furthermore, the spatial heterogeneity in disaster prevention and mitigation capabilities influences the response speed and effectiveness of different regions when dealing with

disasters. Coastal areas, with relatively stronger disaster prevention and mitigation capabilities, are better equipped to withstand disaster impacts.Lastly, the spatial heterogeneity of ecological and environmental conditions has a profound impact on urban sustainable development. Coastal areas should focus on ecological protection and restoration to enhance ecosystem resilience. Inland areas can support sustainable urban development by strengthening ecological conservation.

In summary, the impact of spatial heterogeneity on compound risk distribution is significant. Future disaster – prevention and mitigation work should consider the spatial heterogeneity of different regions, formulate differentiated strategies and measures, and achieve sustainable urban development and effective disaster – risk management.

## 6. Conclusions and recommendations

### 6.1 Research conclusions

**6.1.1 Compound effects significantly amplify flood risks.** Shanghai's inundation risks are driven by the nonlinear compounding of SLR, land subsidence, and storm surges. Under IPCC AR6 projections and ARIMA modeling, the inundation area for SLR alone reaches 361.32 km$^2$ by 2100. With land subsidence added, this area increases to 460.97 km$^2$ (a 28% rise), while incorporating storm surges escalates it to 1,331.91 km$^2$. These findings emphasize that storm surges play a pivotal role in significantly amplifying the risks, especially in the context of cumulative subsidence..

**6.1.2 Spatial heterogeneity shapes regional risk distribution.** The inundation risks in Shanghai demonstrate significant spatial variability, influenced by topography, subsidence rates, and land use patterns. Coastal lowlands (e.g., Chongming, Pudong, Baoshan) face far greater risks than inland areas due to lower elevations and higher subsidence. For instance, by 2100, Chongming's inundation area reaches 725.28 km$^2$, while Pudong's is 393.17 km$^2$. In contrast, inland districts like Jing'an and Huangpu remain largely unaffected.

The uneven spatial distribution of land subsidence exacerbates the vulnerability of coastal areas in Shanghai, especially in densely industrialized and residential zones like Pudong New District. Prolonged subsidence has resulted in lower land elevations, thereby increasing the pressure on flood defenses." In contrast, inland areas, with higher landforms and less subsidence, face smaller direct flood risks, but still need to guard against indirect impacts from extreme rainfall and storm surges, such as urban flooding.Therefore, Shanghai's disaster prevention planning must adopt differentiated strategies based on regional characteristics to effectively tackle the challenges brought by spatial heterogeneity.

**6.1.3 Long-term increase in risks.** Shanghai's inundation risks escalate significantly over the long term (to 2100), particularly under combined SLR, subsidence, and storm surges. The differences between the case considering only SLR and the one combining SLR with land subsidence are minimal in 2050 and 2070, suggesting that the short-term impacts of subsidence are limited. However, by 2100, SLR+land subsidence inundation expands by 28% (460.97 km$^2$ vs. 361.32 km$^2$), highlighting the growing influence of cumulative subsidence. With storm surges added, inundation areas surge to 1,281.07 km$^2$ by 2050 and 1,331.91 km$^2$ by 2100—far exceeding the design capacity of existing defenses (e.g., seawalls at 3.0 meters). These trends underscore the urgent need to enhance infrastructure resilience and emergency preparedness.

### 6.2 Recommendations

**6.2.1 Develop an integrated risk management framework.** To address the multifaceted threats posed by climate change, Shanghai should establish an integrated risk management framework that combines climate adaptation, land subsidence control, and disaster emergency measures. This framework is designed to enhance urban resilience by leveraging data-driven approaches and fostering interagency collaboration. First, the government should invest in a city-wide monitoring network to collect real-time data on SLR, land subsidence, and storm surges, providing a scientific basis for decision – makers. Secondly, an interdepartmental coordination body should be established to integrate resources

from meteorological, water conservancy, planning, and emergency management departments, breaking down information silos and ensuring policy consistency. Then, a risk assessment platform based on big data and AI should be developed. This platform utilizes predictive models to analyze potential flooding areas and disaster impacts, thereby optimizing emergency response plans. Additionally, city – wide risk assessments and actual combat drills should be conducted annually to test and improve the effectiveness of the framework. Through this comprehensive management framework, Shanghai can systematically address compound risks, enhance decision – making efficiency and emergency response capabilities, and maintain urban stability during extreme climate events.

**6.2.2 Implement spatially differentiated adaptation strategies.** Given the varying risk levels across Shanghai, differentiated policies should be implemented to optimize resource allocation. For high-risk areas like Chongming and Pudong, prioritize reinforcing sea defenses, restoring wetlands, and relocating residents to mitigate direct threats from SLR and storm surges. Also, promote sponge city initiatives and upgrade drainage systems to enhance flood prevention. In low-risk areas such as Jing'an and Huangpu, concentrate on disaster education and community drills to boost residents' self – rescue capabilities and ensure swift responses during disasters. Additionally, the government should establish special funds for infrastructure upgrades in high-risk zones and offer tax incentives to encourage corporate participation in disaster prevention projects.

**6.2.3 Promote technological innovation and international collaboration.** Technological innovation is a critical means to enhance Shanghai's climate adaptation capacity. The government should encourage the utilization of remote sensing technology, high-precision digital elevation models (DEMs), and climate modeling technologies to precisely predict inundation extents and risk distribution, providing a scientific basis for urban planning. Concurrently, Shanghai should strengthen collaboration with global coastal cities (e.g., Tokyo, New York) to learn advanced practices in flood control engineering and risk management, while sharing technical resources. Additionally, dedicated research funds should be established to incentivize universities and enterprises to jointly develop adaptive technologies, such as salt-tolerant crops, green building materials, and smart flood control systems.

**6.2.4 Strengthen socioeconomic resilience.** To alleviate the economic and social impacts of climate disasters, Shanghai should adopt a comprehensive strategy encompassing promoting the development of green and low-carbon industries, relocating high-value facilities to lower-risk areas, and establishing robust disaster insurance mechanisms to reduce post-disaster recovery costs. In agricultural areas such as Chongming, promoting climate-resilient agricultural practices like flood-tolerant crops and eco-fisheries can ensure food security and ecological balance. Moreover, enhancing the social security system to provide post – disaster assistance for low – income groups can prevent disasters from worsening social inequalities.Upgrading infrastructure, improving climate monitoring, and refining early warning systems can boost preparedness and response capabilities. Additionally, fostering public awareness and encouraging community involvement in disaster preparedness and response efforts are crucial.

## 7. Research limitations

We selected only the SSP2–4.5 scenario for sea-level rise projections due to its representation of a middle-ground pathway balancing socio-economic development and moderate mitigation efforts. Other SSP scenarios, including SSP1–1.9, SSP1–2.6, SSP3–7.0, and SSP5–8.5, will be explored in future research to cover a broader range of possible futures. In our initial analysis, we added a deterministic subsidence rate to the median SLR projection as a simplified first-order approximation to illustrate the potential compound effect.We acknowledge that this approach does not propagate the full uncertainty inherent in both the SLR projections and the subsidence estimate. Land subsidence prediction is intended to highlight the potential long-term cumulative effect if current subsidence mechanisms persist without mitigation. We fully agree that this approach has limitations. The projection does not account for potential future nonlinearities, changes in groundwater management policy, or other mitigating interventions that could alter the subsidence rate. These predicted values are uncertain values.

## Acknowledgments

Thank Professor Sarah Rogers for reviewing and modifying the manuscript.

## Author contributions

**Conceptualization:** Guoqing Shi.

**Data curation:** Yuexi Wu.

**Investigation:** Yuxuan Zhu.

**Resources:** Zhonggen Sun.

**Supervision:** Mark Wang.

**Writing – original draft:** Bing Liang.

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
