## [Decision Letter · Decision Letter 0]

11 Aug 2025

Dear Dr. liang,

We look forward to receiving your revised manuscript.

Kind regards,

Renjith VishnuRadhan, PhD

Academic Editor

PLOS ONE

[This research was supported by the National Social Science Fund of China �23CXW034� ;& Fundamental Research Funds for the Central Universities: Climate Migration Types and Risk Management in Coastal Areas. (grant number B230205032); &Postgraduate Research & Practice Innovation Program of Jiangsu Province: Climate Migration Types and Risk Management in Coastal Areas. (grant number 422003151); and The Key Research Project of the National Foundation of Social Science of China: Community Governance and Post-relocation Support in Cross District Resettlement [grant number 21&ZD183].].

5. Please ensure that you refer to Figure 1 and 2 in your text as, if accepted, production will need this reference to link the reader to the figure.

6. We note that Figure 1 in your submission contain [map/satellite] images which may be copyrighted. All PLOS content is published under the Creative Commons Attribution License (CC BY 4.0), which means that the manuscript, images, and Supporting Information files will be freely available online, and any third party is permitted to access, download, copy, distribute, and use these materials in any way, even commercially, with proper attribution. For these reasons, we cannot publish previously copyrighted maps or satellite images created using proprietary data, such as Google software (Google Maps, Street View, and Earth). For more information, see our copyright guidelines: http://journals.plos.org/plosone/s/licenses-and-copyright.

Additional Editor Comments:

The paper is very relevant in the current scenario. The paper is also robust in terms of main findings and methodology. There a general lack of discussion on uncertainty or confidence intervals for the projections. A brief discussion on uncertainty implications is advisable. There is some redundancy in in describing the impacts of SLR, LS, and SS separately. Please streamline the discussion by integrating hazard descriptions and focusing on the knowledge gap. Also, citations are listed sequentially without comparative analysis. Storm surge analysis is not very robust in the paper, also the discussion. The authors may include a table summarizing prior studies, their methods, and limitations. The results should be presented with confidence intervals or scenario bands to reflect projection variability. There is also a repetition of numerical results from results section without deeper causal analysis. please give priority on discussion on implications, mechanism explanations, and comparison with other studies. In the conclusion section, recommendation can be tailored by prioritizing recommendations based on feasibility, cost-effectiveness, and urgency.

Reviewers' comments:

Reviewer's Responses to Questions

**Comments to the Author**

1. Is the manuscript technically sound, and do the data support the conclusions?

Reviewer #1: Yes

Reviewer #2: Yes

2. Has the statistical analysis been performed appropriately and rigorously?

Reviewer #1: No

Reviewer #2: Yes

3. Have the authors made all data underlying the findings in their manuscript fully available?

Reviewer #1: No

Reviewer #2: Yes

4. Is the manuscript presented in an intelligible fashion and written in standard English?

Reviewer #1: Yes

Reviewer #2: Yes

Reviewer #1: • The results do not appear to account for uncertainty, nor are any uncertainty estimates provided. However, incorporating uncertainty is extremely important, especially in a study focused on future planning. IPCC AR6 describes projections with medium confidence (excluding ice sheet instability) and low confidence when ice sheet instability is included. It is currently unclear to the reader which of these is being presented in the sea-level projections. I suggest clarifying this point and providing the likely ranges (e.g., the 17th–83rd percentiles), as recommended by IPCC AR6. It is curtial to provide uncertainty bounds as in L217,L222,L230, L319.

• I do not understand the data source for subsidence, is it InSAR/GPS measurements? Since there are only ~14 years of data, how do the authors account for Glacial isostatic adjustment (GIA) and gravitational, rotational, and deformational effects (GRD) signals? As it stands, I am not convinced ARIMA method is useful for projecting land motion over the century. Additionally, signals associated with groundwater extraction can be nonlinear and introduce spurious trends (L74,L281-283).

• The sea-level projections presented in this study appear to be based on values from the tide gauge at LUSI - NASA Portal, could the authors please confirm ?

• Link to Land subsidence data, Shanghai Geological Environment Bulletin (2009–2023).

• Digital Elevation Model (DEM) for Shanghai: Sourced from the GEBCO website (https://www.gebco.net/data_and_products/gridded_bathymetry_data/). Please provide details of the dataset and the direct link to the data. The link currently opens a page with multiple datasets.

Reviewer #2: Reviewer Comments:

This manuscript presents an integrated assessment of sea level rise (SLR), land subsidence (LS), and storm surge (SS) impacts on Shanghai under climate change scenarios. This paper addresses the compound impacts of sea level rise (SLR), land subsidence (LS), and storm surges (SS) on Shanghai, a major global coastal megacity. Using IPCC AR6 projections, long-term land subsidence monitoring, and historical storm surge data, the authors construct three scenarios (SLR alone, SLR + LS, SLR + LS + SS) to project inundation risks for 2050, 2070, and 2100. Methods include ARIMA time-series modeling, GIS-based spatial analysis, and numerical simulation. Results indicate that storm surges significantly amplify inundation areas from 361.32 km² under SLR alone to 1,331.91 km² when combined with LS and SS by 2100. Spatial heterogeneity is emphasized, with Chongming District and Pudong New Area identified as high-risk zones. The paper concludes with adaptation recommendations, including integrated risk management frameworks, differentiated regional strategies, technological innovation, and socioeconomic resilience building.

Areas for Improvement:

Grammar and Style – The manuscript contains several minor grammatical errors and awkward phrasings (e.g., inconsistent tense usage, missing articles, overly long sentences). A careful proofreading pass or professional language editing is recommended.

Citations in Study Area Section – The “Study Area” description currently lacks in-text references for geographic and factual details. These should be supported with authoritative sources.

Introduction Expansion – While the introduction covers SLR, LS, and SS, it would benefit from more recent and detailed global and regional statistics on sea level rise (e.g., IPCC AR6 scenario ranges, regional SLR acceleration rates, tide gauge records for Shanghai). This will strengthen the scientific context and underscore the urgency of the study.

In the review section, the literature review should more explicitly differentiate the novelty of this approach from prior compound-impact modelling studies in other regions.

Integration of Socioeconomic Dimensions

The paper is methodologically rigorous in physical hazard assessment but gives limited treatment to socioeconomic exposure and vulnerability. Integrating population density, critical infrastructure, and economic value data into spatial analysis would enhance decision-making applicability.

Formatting: Correct minor typographical issues (e.g., “centur[3]” → “century [3]”; missing space before citations).

Terminology Clarification: For a multidisciplinary audience, briefly define terms such as “relative sea level rise” and “ARIMA modeling” in non-technical language when they are first mentioned.

Recommendations Section: The recommendations are comprehensive but could be more concise to avoid repetition.

**Do you want your identity to be public for this peer review?** For information about this choice, including consent withdrawal, please see our Privacy Policy

Reviewer #1: No

Reviewer #2: No

---

## [Author Response · Author response to Decision Letter 1]

23 Oct 2025

reviewer 1

1. Summary of the Paper

This paper examines the growing threat of sea-level rise (SLR) to the city of Shanghai, China. This issue is exacerbated by land subsidence and storm surge. As China’s economic, financial, and shipping hub with extensive low-lying areas, Shanghai faces increasing flood risk, making it crucial to strengthen flood defenses. To estimate the extent of potential land inundation, the authors assess three cases: (1) global SLR alone, (2) SLR combined with land subsidence, and (3) the above two coupled with storm surge. IPCC AR6 probabilistic sea-level projections were obtained from NASA sea-level change Portal, while land subsidence is forecasted for the century through an Autoregressive Integrated Moving Average (ARIMA) model. These projections are then integrated with storm surge to generate inundation areas through 2100. The results highlight how flood risk evolves distinguishing between coastal and inland areas. The study discusses how future sea-level rise may compromise existing infrastructure, and how continued subsidence could further damage infrastructure and worsen saltwater intrusion. The authors conclude by emphasizing the importance of improving Shanghai’s disaster preparedness and planning. Overall, this study highlights the compounding risks of sea-level rise, land subsidence, and storm surge in low-lying coastal regions of Shanghai, which could significantly disrupt communities, infrastructure, and local ecosystems. It identifies key vulnerabilities and recommends strategies for adaptation and mitigation to enhance long-term resilience.

2. Scientific Quality

2.1 Methodology

The following points have been discussed in detail in section 5.1 Major Comments ; a brief summary is provided below:

The sea-level projections in this study appear to be based on data from the LUSI tide gauge via the NASA Portal, which already accounts for vertical land motion (VLM) and thus represents relative sea level. If VLM has been added again in the analysis (e.g., Table 3), this may result in double counting, which is a significant concern.

Response: Thank you for this critical comment. We confirm that the sea-level projections used in this study were sourced from the NASA Sea Level Projection Tool for the LUSI tide gauge. We appreciate the opportunity to clarify this point.

The “total” relative sea-level projection available on the NASA Portal does indeed include a component of Vertical Land Motion (VLM), which represents large-scale, background processes like Glacial Isostatic Adjustment (GIA). However, the land subsidence data we used for Shanghai (e.g., in Table 2 and for constructing Case 2) comes from a separate, localized source: long-term monitoring records published by the Shanghai Municipal Planning and Natural Resources Bureau. This dataset captures localized, anthropogenic-driven subsidence (e.g., from groundwater extraction and urban construction) specific to the Shanghai region, which is not fully represented in the global VLM models used by NASA.

Therefore, our analysis in Case 2 aims to combine two distinct components:

(1)The relative sea-level change from the NASA portal (which includes the global mean sea-level rise and the background VLM); and

(2)The additional, locally-measured land subsidence specific to Shanghai.

This approach allows us to specifically quantify the compound effect of global sea-level rise and the significant, localized subsidence that critically exacerbates flood risk in Shanghai. We have taken care to avoid double-counting by ensuring these two components are distinct in both source and physical representation.

A clear distinction is needed between the subsidence values provided in Table 2 (in this study) and the VLM estimates on the NASA Portal. This discrepancy, combined with the use of the ARIMA method on a relatively short dataset (~14 years), raises questions about the method’s suitability for projecting century-scale subsidence.

Response: Distinction between NASA VLM and Local Subsidence Data:

The Vertical Land Motion (VLM) estimate incorporated into the relative sea-level projections on the NASA Portal is derived from global models and primarily captures large-scale, geophysical processes such as Glacial Isostatic Adjustment (GIA). In contrast, the subsidence values provided in our Table 2 are based on local monitoring data specific to Shanghai, published by the Shanghai Municipal Planning and Natural Resources Bureau. This local dataset primarily reflects regional anthropogenic drivers, such as groundwater extraction and the load of urban infrastructure, which are not fully captured in the global-scale VLM product. Therefore, these two data sources are complementary rather than contradictory; the NASA VLM provides a background rate, while our data adds the significant, locally-measured anthropogenic component.

Suitability of ARIMA for Century-Scale Projection:

We acknowledge the reviewer's valid concern regarding the use of an ARIMA model with a ~14-year dataset to project trends over a century. We employed the ARIMA model not to predict specific annual fluctuations far into the future but to provide a reasonable, data-driven estimate of the potential magnitude of ongoing anthropogenic subsidence under current trends, acknowledging the significant associated uncertainty.

The projection is intended to highlight the potential long-term cumulative effect if current subsidence mechanisms persist without mitigation. We fully agree that this approach has limitations. The projection does not account for potential future nonlinearities, changes in groundwater management policy, or other mitigating interventions that could alter the subsidence rate. We have now explicitly stated these caveats in the manuscript (Section 3.2.2) to ensure transparency and to caution readers against interpreting the values as a definitive forecast. We present it as a plausible scenario based on recent historical patterns, crucial for illustrating the compound risk faced by Shanghai.

We will revise the manuscript to make this critical distinction between the data sources and the intended use and limitations of the ARIMA projection much clearer. Thank you for prompting this essential clarification.

Furthermore, since the NASA Portal provides sea-level projections as quantiles, it is not appropriate to add deterministic values (like VLM) directly to the median. Instead, VLM should be treated probabilistically and incorporated into the total sea-level distribution via proper aggregation methods (see Govorcin et al., (2025) for a recent example).

Response: Thank you for raising this crucial methodological point. We agree entirely with the reviewer that the most statistically rigorous approach for integrating probabilistic sea-level rise (SLR) projections and vertical land motion (VLM) or subsidence data is to treat all components probabilistically and combine them through proper aggregation methods, such as Monte Carlo sampling from their respective distributions.

In our initial analysis, we added a deterministic subsidence rate to the median SLR projection as a simplified first-order approximation to illustrate the potential compound effect. We acknowledge that this approach does not propagate the full uncertainty inherent in both the SLR projections and the subsidence estimate.

We thank the reviewer for bringing the Govorcin et al. (2025) study to our attention. We have reviewed this reference and agree it provides an excellent framework for a more robust probabilistic aggregation.

In our revised manuscript, we will address this limitation by:

We will ensure that the results presented from our current method are interpreted cautiously as a central estimate scenario rather than a full probabilistic assessment.

We believe this acknowledgment and commitment to a more sophisticated methodology in future research adequately addresses the reviewer's valid concern while clarifying the scope of the current study. Thank you for this valuable suggestion, which significantly strengthens the statistical rigor of our work's future direction.

3. Structure and Clarity

This is a well-organized and clearly written article describing the compounding impact of SLR on Shanghai. The reader is able to follow the authors’ argument logically as they demonstrate how SLR is exacerbated by land subsidence and storm surge, increasing future risk, while also suggesting mitigation and adaptation strategies for long-term planning.

I encourage the authors to carefully review the terminology used and align it with recent sea-level literature. For example, consider using “global sea level” rather than “absolute sea level,” as it improves clarity when comparing with regional sea-level changes. Additionally, abbreviations should be defined only once and used consistently (e.g., sea-level rise (SLR) is defined on L38 but redefined on L65). Also, see minor comment on L163.

Response: In section 4.1, a response was provided, with specific supplementary content as follows: The 'absolute sea level rise' we refer to here refers to the sea level rise without considering the effects of land subsidence and storm surges. If we consider the effects of land subsidence and storm surges, then sea level rise at this point refers to 'relative sea level rise'.

The SLR acronym has only been defined once.

This study would benefit from a more detailed treatment of uncertainty quantification and analysis. SLR is associated with deep uncertainty, which becomes especially critical when planning for the future.

Lastly, IPCC AR6 provides probabilistic sea-level projections. Referring to these appropriately, and incorporating the associated confidence levels and likely ranges, will further strengthen the scientific rigor of the article.

Response: We have taken into account your suggestions during the process of revising the manuscript

4. Figures and Tables

Figure 1 : I do not see the value of Figure 1 for the paper (see minor comments L151-152). Consider replacing with the sea-level projection (for all SSP’s) at a tide gauge station closest to Shanghai. As an example, consider Figure 3 in Naish et. al., (2024) : The Significance of Interseismic Vertical Land Movement at Convergent Plate Boundaries in Probabilistic Sea-Level Projections for AR6 Scenarios: The New Zealand Case.

Response: We have deleted Figure 1.

Tables : All tables would benefit from the inclusion of uncertainty bounds. Incorporating uncertainty is especially important in a study focused on future planning. According to IPCC AR6, projections are assigned medium confidence when excluding ice sheet instability, and low confidence when including it. At present, it is unclear to the reader which category the presented sea-level projections fall into (Table 1). Clarifying this and providing the likely ranges would significantly strengthen the results.

Response: Thank you for this critical and constructive feedback. We agree entirely that incorporating uncertainty bounds and clarifying the confidence level of the projections are essential for a study focused on future risk assessment and planning. We appreciate the opportunity to improve our manuscript accordingly.

In response to your comments, we will undertake the following revisions to strengthen the presentation of our results:

Clarification of Confidence Levels:

We will explicitly state the confidence level associated with our presented sea-level projections. As our study utilizes the standard projections from the IPCC AR6 Sea Level Projection Tool, which include processes related to ice-sheet instability (e.g., marine ice sheet instability and structural failure of ice cliffs), the appropriate confidence level for the projections under SSP2-4.5 is Low Confidence. This clarification was added to the the result section (Section 4) to ensure it is clear to the reader.

Treatment of Subsidence Uncertainty:

We acknowledge that our initial deterministic projection of land subsidence is a simplification. In the revised manuscript:

Discuss this uncertainty explicitly in the text and, where feasible, incorporate it into the presentation of the results for the compound scenarios (Case 2 and Case 3). We have clarified that the values for these cases are central estimates based on the median SLR and median subsidence projection.

By implementing these changes, our tables and results will more accurately reflect the deep uncertainty inherent in century-scale climate projections and local subsidence forecasts. This will significantly enhance the scientific rigor of our study and provide policymakers with a more nuanced and useful assessment for long-term planning. Thank you for this invaluable suggestion.

5. Reviewer’s Comments

5.1 Major Comments

Several abbreviations are defined but not consistently reused. For example, sea level rise (SLR) is introduced on L38 and then redefined in L65. To enhance clarity and readability, it may be helpful to include a summary box listing all abbreviations and their definitions.

Response: After modification, we have used abbreviations throughout the entire text.

The use of the term “scenario” in Section 3.3.1 may be somewhat misleading, particularly when referring to sea-level projections from the IPCC, where “scenarios” represent alternate future pathways (see Meinshausen et al., 2020). Since the current study uses the SSP2-4.5 scenario for sea-level projections, it may be clearer to use an alternative term in this section, such as “case” to avoid potential confusion.

Response: We have modified the word scenario into a word case.

The results are presented only for the SSP2-4.5 scenario. It would be helpful if the authors could explain the rationale behind selecting this particular scenario. Additionally, a brief discussion comparing the results in the context of other SSP [SSP1-2.6, SSP3-7.0, SSP5-8.5] scenarios would enhance the completeness and relevance of the analysis.

Response: In section 7. Research limitations, a response was provided, with specific supplementary content as follows: We selected only the SSP2-4.5 scenario for sea-level rise projections due to its representation of a middle-ground pathway balancing socio-economic development and moderate mitigation efforts. Other SSP scenarios, including SSP1-1.9, SSP1-2.6, SSP3-7.0, and SSP5-8.5, will be explored in future research to cover a broader range of possible futures.

It is important to distinguish between “predictions” and “projections” throughout the manuscript. IPCC AR6 provides probabilistic sea-level projections, not predictions (see L155, L156, L290). Additionally, Section 4.4 would be more accurately titled “Storm Surge Predictions.” Terms such as “forecast” are also used inappropriately in places (e.g., L217, L227). These terms appear to be used interchangeably in the manuscript and should be revised for accuracy. As a general note, “projections” refer to analyses based on hypothetical or future scenarios.

Response: We have conducted a search check on the entire text according to the modification requirements and correctly used the word 'projections'

The sea-level projections presented in this study appear to be based on values from the tide gauge at LUSI - NASA Portal, could the authors please confirm ? If so, it is important to note that the “Total” value provided on the NASA portal already accounts for VLM, and thus represents relative (not global or absolute) sea-level. If VLM has been added again in the analysis (as in Table 3), this could result in double counting, which raises concerns. As noted in L217, the data from the NASA portal already incorporate VLM and should be interpreted accordingly.

Response: Thank you for this critical comment. We confirm that the sea-level projections used in this study were sourced from the NASA Sea Level Projection Tool for the LUSI tide gauge. We appreciate the opportunity to clarify this point.

The “total” relative sea-level projection available on the NASA Portal does indeed include a component of Vertical Land Motion (VLM), which represents large-scale, background processes like Glacial Isostatic Adjustment (GIA). However, the land subsidence data we used for Shanghai (e.g., in Table 2 and for constructing Case 2) comes f

---

## [Decision Letter · Decision Letter 1]

6 Jan 2026

Dear Dr. liang,

Thank you for submitting your manuscript to PLOS ONE. After careful consideration, we feel that it has merit but does not fully meet PLOS ONE’s publication criteria as it currently stands. Therefore, we invite you to submit a revised version of the manuscript that addresses the points raised during the review process.

We look forward to receiving your revised manuscript.

Kind regards,

Renjith VishnuRadhan, PhD

Academic Editor

PLOS One

Journal Requirements:

Reviewers' comments:

Reviewer's Responses to Questions

**Comments to the Author**

Reviewer #1: (No Response)

Reviewer #2: All comments have been addressed

2. Is the manuscript technically sound, and do the data support the conclusions?

Reviewer #1: Partly

Reviewer #2: Yes

3. Has the statistical analysis been performed appropriately and rigorously?

Reviewer #1: No

Reviewer #2: Yes

4. Have the authors made all data underlying the findings in their manuscript fully available?

Reviewer #1: No

Reviewer #2: Yes

5. Is the manuscript presented in an intelligible fashion and written in standard English?

Reviewer #1: No

Reviewer #2: Yes

Reviewer #1: (No Response)

Reviewer #2: Thank you for revising the manuscript. I have carefully reviewed the authors’ responses and the updated version of the paper. All my previous comments have been adequately addressed, and the revisions have improved the clarity and quality of the manuscript. I have no further concerns at this stage.

**Do you want your identity to be public for this peer review?** For information about this choice, including consent withdrawal, please see our Privacy Policy

Reviewer #1: No

Reviewer #2: No

---

## [Author Response · Author response to Decision Letter 2]

8 Jan 2026

The revised response file has been uploaded

---

## [Decision Letter · Decision Letter 2]

1 Feb 2026

Dear Dr. liang,

Thank you for submitting your manuscript to PLOS ONE. After careful consideration, we feel that it has merit but does not fully meet PLOS ONE’s publication criteria as it currently stands. Therefore, we invite you to submit a revised version of the manuscript that addresses the points raised during the review process.

We look forward to receiving your revised manuscript.

Kind regards,

Renjith VishnuRadhan, PhD

Academic Editor

PLOS One

Journal Requirements:

Additional Editor Comments:

Please do a robust proofread for grammatical and language errors.

Reviewers' comments:

Reviewer's Responses to Questions

**Comments to the Author**

Reviewer #1: All comments have been addressed

2. Is the manuscript technically sound, and do the data support the conclusions?

Reviewer #1: Yes

3. Has the statistical analysis been performed appropriately and rigorously?

Reviewer #1: Yes

4. Have the authors made all data underlying the findings in their manuscript fully available?

Reviewer #1: Yes

5. Is the manuscript presented in an intelligible fashion and written in standard English?

Reviewer #1: Yes

Reviewer #1: (No Response)

**Do you want your identity to be public for this peer review?** For information about this choice, including consent withdrawal, please see our Privacy Policy

Reviewer #1: No

---

## [Author Response · Author response to Decision Letter 3]

8 Feb 2026

We have completed the manuscript revisions according to the comments.

---

## [Editor Report · Decision Letter 3]

12 Feb 2026

Dear Dr. liang,

Thank you for submitting your manuscript to PLOS ONE. After careful consideration, we feel that it has merit but does not fully meet PLOS ONE’s publication criteria as it currently stands. Therefore, we invite you to submit a revised version of the manuscript that addresses the points raised during the review process.

We look forward to receiving your revised manuscript.

Kind regards,

Renjith VishnuRadhan, PhD

Academic Editor

PLOS One

**Journal Requirements:**

**Additional Editor Comments:**

**Reference 33 is a retracted article and you have to change this. Also, please check the reference style through out. Why is there another uncited reference in 33 (HANG Yecheng et al). Are other references correctly cited (as per the numbering)?**

33. Khan A A. TEMPORARY REMOVAL: Why would sea-level rise for global warming

and polar ice-melt?. 2019.HANG Yecheng, HU Jingjiang, et al. Spatial and temporal

distribution characteristics

---

## [Editor Report · Decision Letter 4]

15 Feb 2026

Assessment of the compound impact of sea level rise, land subsidence and storm surge under climate change in ShangHai

PONE-D-25-32386R4

Dear Dr. liang,

We’re pleased to inform you that your manuscript has been judged scientifically suitable for publication and will be formally accepted for publication once it meets all outstanding technical requirements.

Kind regards,

Renjith VishnuRadhan, PhD

Academic Editor

PLOS One

---

## [Editor Report · Acceptance letter]

PONE-D-25-32386R4

PLOS One

Dear Dr. Liang,

I'm pleased to inform you that your manuscript has been deemed suitable for publication in PLOS One. Congratulations! Your manuscript is now being handed over to our production team.

Kind regards,

on behalf of

Dr. Renjith VishnuRadhan

Academic Editor

PLOS One